# Treatment of Diet-Induced Obese Rats with CB_2_ Agonist AM1241 or CB_2_ Antagonist AM630 Reduces Leptin and Alters Thermogenic mRNA in Adipose Tissue

**DOI:** 10.3390/ijms24087601

**Published:** 2023-04-20

**Authors:** Lannie O’Keefe, Teresa Vu, Anna C. Simcocks, Kayte A. Jenkin, Michael L. Mathai, Andrew J. McAinch, Dana S. Hutchinson, Deanne H. Hryciw

**Affiliations:** 1Institute for Health and Sport, Victoria University, P.O. Box 14428, Melbourne, VIC 8001, Australia; 2Drug Discovery Biology, Monash Institute of Pharmaceutical Sciences, Monash University, Parkville, VIC 3052, Australia; 3School of Science, Western Sydney University, Campbelltown, NSW 2560, Australia; 4The Florey Institute of Neuroscience and Mental Health, Parkville, VIC 3052, Australia; 5Australian Institute for Musculoskeletal Science (AIMSS), Victoria University, Melbourne, VIC 8001, Australia; 6School of Environment and Sciences, Griffith University, Nathan, QLD 4111, Australia; 7Griffith Institute for Drug Discovery, Griffith University, Nathan, QLD 4111, Australia

**Keywords:** diet-induced obesity (DIO), inflammation, endocannabinoid system (ECS), AM1241, AM630, adipose tissue, cannabinoid receptor 2 (CB_2_)

## Abstract

Diet-induced obesity (DIO) is a contributor to co-morbidities, resulting in alterations in hormones, lipids, and low-grade inflammation, with the cannabinoid type 2 receptor (CB_2_) contributing to the inflammatory response. The effects of modulating CB_2_ with pharmacological treatments on inflammation and adaptations to the obese state are not known. Therefore, we aimed to investigate the molecular mechanisms in adipose tissue of CB_2_ agonism and CB_2_ antagonism treatment in a DIO model. Male Sprague Dawley rats were placed on a high-fat diet (HFD) (21% fat) for 9 weeks, then received daily intraperitoneal injections with a vehicle, AM630 (0.3 mg/kg), or AM1241 (3 mg/kg), for a further 6 weeks. AM630 or AM1241 treatment in DIO rats did not alter their body weight, food intake, or liver weight, and it had no effect on their numerous circulating cytokines or peri-renal fat pad mass. AM1241 decreased heart weight and BAT weight; both treatments (AM630 or AM1241) decreased plasma leptin levels, while AM630 also decreased plasma ghrelin and GLP-1 levels. Both treatments decreased Adrb3 and TNF-α mRNA levels in eWAT and TNF-α levels in pWAT. AM630 treatment also decreased the mRNA levels of Cnr2, leptin, and Slc2a4 in eWAT. In BAT, both treatments decreased leptin, UCP1, and Slc2a4 mRNA levels, with AM1241 also decreasing Adrb3, IL1β, and PRDM16 mRNA levels, and AM630 increasing IL6 mRNA levels. In DIO, CB_2_ agonist and CB_2_ antagonist treatment reduces circulating leptin in the absence of weight loss and modulates the mRNA responsible for thermogenesis.

## 1. Introduction

The endocannabinoid system regulates multiple metabolic processes, with endogenous endocannabinoid ligands likely to play an important role in appetite regulation, energy balance and metabolism, thermoregulation, and immunological functions [1,2]. This makes targeting the endocannabinoid system attractive from a therapeutic standpoint in the treatment and management of obesity and its co-morbidities. At this time, there is limited understanding of the metabolic pathways that are responsible for the co-morbidities associated with obesity, such as systemic inflammation, which impacts the ability to develop effective therapeutics. A clear understanding of the endocannabinoid pathway in disease pathologies is warranted, especially given the increased use of cannabinoids as a medical treatment worldwide [3].

There are two cannabinoid receptors: cannabinoid receptor 1 (CB_1_) and cannabinoid receptor 2 (CB_2_). The CB_2_ receptor shares only 44% amino acid homology with the CB_1_ receptor, and they have varying metabolic roles which may, in part, be due to the differences in their tissue distribution [4]. While there is extensive research on the role of CB_1_ receptors and their potential use in the treatment of obesity [2,5], the role of CB_2_ in obesity is less clear. We have previously demonstrated that the consumption of a high-fat diet (HFD) reduces the renal CB_2_ protein [6]. Further, in a mouse model of obesity, the CB_2_ agonist treatment increased the cell size of the epididymal fat [5] and reduced the kidney size [6]. Agonism of CB_2_ with JWH-015 reduced body weight and food intake [5], while there was no effect on body weight for two other CB_2_ agonists, AM1241 or JWH-133 [6,7]. CB_2_-knockout mice fed an HFD displayed a reduced body weight gain in one study [7]; however, another study showed no differences in weight gain compared to their wildtype counterparts [8]. Despite this, CB_2_-knockout mice that were fed a chow diet became obese and displayed an increase in proinflammatory markers [9], and the pharmacological activation of the CB_2_ receptor reduced inflammation (as reviewed in [10]). As CB_2_ appears to be downregulated in obesity [6], this has led to some researchers hypothesizing that the activation of the CB_2_ receptor could help alleviate obesity-driven inflammation. In the obese state, there is a prolonged increase in cytokine levels, which correlates with fat storage [11]. Adipose tissue is dysregulated in obesity, and there are conflicting reports on the importance of CB_2_ receptors in adipose tissue. CB_2_ is expressed in adipose tissue in the stromal vascular fraction as opposed to adipocytes themselves, and CB_2_ mRNA expression is upregulated in HFD-fed mice and *ob/ob* mice with adipose tissue pads [7]. There are conflicting reports on the effect of an HFD on the adipocyte size in CB_2_-knockout mice: one study showed a similar epididymal white adipocyte size compared to the wildtype mice [8], while another study showed a hypertrophy of visceral fat [9]. The treatment of HFD mice with the CB_2_ agonist JWH-133 potentiated fat inflammation while treatment with the CB_2_ antagonist decreased fat inflammation [7], which is consistent with the reduced inflammatory gene expression in CB_2_-knockout mice on an HFD [7,12].

Therefore, in this study, we investigated the effects of chronic treatment with the CB_2_ agonist AM1241 or the CB_2_ antagonist AM630 in DIO rats [6] and the impact these compounds have on (1) plasma inflammatory markers, (2) the mRNA expression of inflammatory markers in adipose tissue, and (3) specific adipocyte genes involved in the browning of adipose tissue.

## 2. Results

### 2.1. AM1241 or AM630 Treatment of DIO Rats Alters Body Adiposity

Male Sprague Dawley rats were placed on an HFD for 9 weeks and then allocated to either a vehicle, AM1241, or AM630 treatment for a further 6 weeks [13]. While AM1241 treatment did not alter their body weight or food intake (as we have previously reported [6]), there was a significant decrease in the amount of epididymal white adipose tissue as a percentage of the total body weight (*p* = 0.052 for absolute tissue weight), and there was a trend for decreased peri-renal adipose tissue weight (*p* = 0.052) and lean mass in grams to increase (*p* = 0.08). Additionally, treatment with AM630 did not alter their body weight or food intake [6], but it significantly increased the lean mass in grams and reduced epididymal white adipose tissue and brown adipose tissue weights (Table 1; *p* < 0.05) and trended to increase liver weight (*p* = 0.058). Both the AM1241 and AM630 treatments of DIO rats caused a significant increase in heart weights (Table 1; *p* < 0.05).

### 2.2. AM1241 and AM630 Treatment of DIO Rats Lowers Plasma Leptin

We have previously demonstrated that AM1241 and AM630 treatment in DIO rats has no impact on glucose or insulin tolerance [6]. In this study, AM1241 in DIO rats did not alter plasma levels of adiponectin, ghrelin, glucagon, PAI-1, or GLP-1 (Figure 1a,b,d–f), but it significantly decreased leptin compared to DIO vehicle-treated rats (Figure 1c; *p* < 0.05). While AM630 treatment significantly decreased leptin levels, it also significantly decreased plasma ghrelin and GLP-1 levels (Figure 1b,c,f; *p* < 0.05).

### 2.3. AM1241 or AM630 Treatment of HFD Rats Fails to Alter Plasma Cytokines

We measured circulating cytokine inflammatory markers to assess whether AM1241 or AM630 treatment in DIO could alter these. Compared to the vehicle-treated rats, AM1241 or AM630 had no effect on any of the cytokine markers measured in DIO (Table 2).

### 2.4. Effect of AM1241 in DIO on Adipose Tissue Gene Expression

With varying effects of AM1241 and AM630 on adipose tissue depot weights (Table 1), we then investigated the mRNA expression of the genes involved in browning and brown adipocyte markers (*UCP1, PRDM16*, and *CPT1b*), the white adipocyte markers *HOXC9* and *TCF21*, inflammatory mediators (IL-1β, IL-6, and TNF-α), the adipokines adiponectin and leptin, glucose transporters (*Slc2a1* and *Slc2a4*), the *β*_3_*-AR (Adrb3),* which is important for adipocyte thermogenesis as well as for the expression of the *Cnr1* and *Cnr2* receptors.

In epididymal WAT (eWAT), AM1241 treatment in DIO rats caused a significant decrease in *Adrb3* and *PRDM16* (Figure 2; *p* < 0.05), with no changes in any of the other genes investigated. AM630 treatment in DIO rats also caused a significant decrease in *Adrb3* and PRDM16 mRNA levels, and significantly decreased *leptin, Slc2a4*, and *Cnr2* mRNA levels, with a non-significant trend for a reduction in adiponectin (*p* = 0.051), *CPT1b* (*p* = 0.053), and *HOXC9* (*p* = 0.051).

In peri-renal WAT (pWAT), AM1241 treatment in DIO rats caused a significant decrease in *CPT1B* and *TNF-α* (Figure 3; *p* < 0.05). Whereas a decrease in *Cnr1, Cnr2*, and *TNF-α* (Figure 3; *p* < 0.05) and a non-significant trend for a reduction in adiponectin (*p* = 0.053) were observed following six weeks of treatment with AM630.

In BAT, AM1241 or AM630 treatment of DIO rats caused a significant decrease in the mRNA expression of *leptin, Slc2a4*, and *UCP1*, whereas *Adrb3, IL-1β,* and *PRDM16* were only decreased with AM1241, and *IL-6* mRNA expression increased only after AM630 treatment (Figure 4; *p* < 0.05). *Cnr1* and *Cnr2* were undetectable in BAT.

## 3. Discussion

Endocannabinoid system dysfunction is observed in obesity, but the mechanisms by which this contributes to the etiology and the associated co-morbidities are not yet fully understood [14]. With discrepancies in the role of the CB_2_ in obesity and adipose tissue function and inflammation, we aimed to investigate the effect of either chronic CB_2_ agonism with AM1241 or chronic CB_2_ antagonism with AM630 on inflammation, and adipose tissue in a DIO rat model. AM1241 or AM630 treatment of DIO rats fed an HFD diet failed to alter their body weight or food intake [6]. Surprisingly, both treatments reduced eWAT mass, while AM630 also reduced BAT weight, with a non-significant trend for AM1241 to reduce pWAT weights. The effects on reduced eWAT mass in DIO treated with either AM1241 or AM630 were associated with a decrease in plasma leptin concentrations, reduced leptin mRNA levels in BAT, and reduced leptin mRNA levels in eWAT (AM630 treatment only). Leptin, an adipokine, regulates whole-body energy balance through neuronal appetite pathways as well as a number of peripheral effects leading to decreased adipocyte fat storage [15]. In white adipocytes derived from obese patients, the leptin mRNA levels decreased following CB_2_ stimulation with JWH-133 [16], whereas in CB_2_-knockout mice following an HFD for a year, increased levels of leptin that correlated with adipose storage were displayed [12]. While we observed no changes in food intake with either AM1241 or AM630 levels, these combined results may implicate a protective role of leptin in conjunction with the CB_2_ receptor that is independent of body weight/food intake. The effects of AM630 are more complicated, as the plasma concentrations of ghrelin and GLP-1 were also reduced at the same time.

In our DIO models, we observed differences in the effect of AM1241 on body weight/food intake between this study and JWH-015 in our previous study. There are several factors that could contribute to this discrepancy, including the following: (1) AM1241 is a partial agonist with respect to cAMP assays at the human CB_2_ receptor, whereas, JWH-015 is a full agonist [5]; (2) receptor selectivity (AM1241 displays more CB_2_/CB_1_ binding selectivity compared to JWH-015) [17]; (3) species difference (while JWH-015 is a full agonist at cAMP assays at the human and mouse receptors, AM1241 is an inverse agonist at the mouse CB_2_ receptor and a partial agonist at the human receptor [17]; and (4) the number and variety of off-target effects by both drugs [17]. All these factors may contribute to the differences observed in this study and our previous study.

There is widespread inflammation in obesity (including in animal models), but there is little information on whether CB_2_ activation or inactivation can improve inflammation in obese rodents. CB_2_ activation by either CB_2_ selective agonists or CB_1_/CB_2_ agonists in the presence of CB_1_ antagonists improves inflammation in a range of inflammatory models (as reviewed in [9]). In the current study, we have demonstrated that *TNF-α* mRNA expression was reduced in peri-renal WAT, which is in agreement with our previous observations that 21 days of treatment with JWH-015 reduces TNF-α in retroperitoneal WAT [5]. Similarly, 21 days of treatment with the CB_2_ agonist JWH-133 in obese mice results in a decrease in the pro-inflammatory biomarkers *TNF-α, IL-6,* and *IL-β* [18]. Intriguingly, obese fat-fed *ob/ob* mice treated with CB_2_ agonist JWH-133 for 15 days showed an increase in CB_2_ expressions in the stromal vascular fraction of eWAT in correlation with increased adipose tissue inflammation [7]. However, following chronic activation of the CB_2_ receptor with AM1241 in the current study, we did not observe any alterations in a wide range of plasma cytokines, or the expression of *IL-6* in any of the adipose tissue depots investigated, or TNF-α in any of the other adipose tissue depots. Overall, this suggests that CB_2_ agonism does not improve obesity-driven inflammation. Chronic treatment with the CB_2_ antagonist AM630 also reduced *TNF-α* mRNA levels in pWAT and increased *IL-6* mRNA levels in BAT. These alterations did not result in changes in any inflammatory plasma concentration measured. This is in contrast to a previous study that showed that AM630 reduced fat inflammation [7]. This discrepancy may be due to differences in the dose used (1 mg/kg/day for 15 days) or the fact that those experiments were performed in ob/ob mice (note that AM630 had no effect in the lean *ob+/ob* mice, further supporting a link between leptin and CB_2_ receptors). It should be noted that while there is evidence that CB_2_ is expressed in the stromal vascular fraction of WAT and not in mature adipocytes themselves [7], an HFD increases CB_2_ expression in eWAT. In our study, AM630 decreased *CB*_2_ mRNA levels in eWAT and pWAT (with a trend for AM1241 to do the same in eWAT). We measured mRNA in tissue depots, and it may be worthwhile in the future to examine changes in mRNA levels in the SVF instead, where the CB_2_ receptor is expressed.

We observed a decrease in the epididymal fat pad as a percentage of body weight following AM1241 treatment. A direct correlation between low leptin levels and a decrease in the epididymal fat pad mass has been previously observed in rats [19]. In mice, epididymal fat has been found to initiate leptin control via afferent nerve signaling to control appetite. The increased expression of *UCP1* in epididymal fat reduced appetite via afferent nerve signaling [20]. Our results found that CB_2_ activation and inhibition reduced epidydimal fat expression of *UCP1*, which is consistent with the lack of effect on appetite that we observed [6]. The reduction in the epididymal fat pad mass and the low circulating concentrations of leptin detected imply that our results further depict a protective role of leptin working in conjunction with CB_2_ independent of affecting body weight and food consumption. In agreement with this, our results found that both the activation and inhibition of CB_2_ reduced the expression of *UCP1* in epididymal adipose, without any changes in food intake. BAT is an important organ for controlling whole-body energy homeostasis through its roles in thermoregulation and glucose uptake [21]. While there is some evidence that CB_1_ antagonism may be beneficial for increased the BAT activity in rodents and humans [22], there is little research on CB_2_ and its impact on BAT function. For example, *UCP1* mRNA in BAT in CB_2_-knockout mice is unaltered [12] and JWH-015-treated HFD mice show no changes in the *UCP1* protein [5]. *Cnr1* and *Cnr2* mRNAs were not detected in BAT; hence, the effects of AM630 and AM1241 on the transcriptional changes in BAT are most likely due to a secondary indirect effect [23]. Since BAT is highly innervated, this could be due to alterations in the sympathetic tone to BAT. Another possible mechanism is the reduction in circulating leptin levels following both AM630 and AM1241 treatment. Leptin administration to rats increases BAT *UCP1* mRNA levels, which are dependent upon the sympathetic activation of BAT [23]. While we show that neither the *CB*_1_ nor *CB*_2_ receptors were detected in BAT, the treatment of DIO rats with AM1241 reduces *UCP1*, *Adrb3*, *Slc2a4*, and *PRDM16* mRNA without affecting the BAT mass.

UCP1 is a mitochondrial protein that is responsible for non-shivering thermogenesis and can be activated by the neurotransmitter noradrenaline, which acts at BAT *β*_3_*-ARs* to increase thermogenesis [21]. The reduced expression of both *Adrb3* and *UCP1*, and the transcriptional co-regulator *PRDM16*, which controls the development of brown adipocytes in BAT, may suggest a blunting of thermogenic responses in these animals. *Adrb3* is a regulator of CB_2_ thermogenesis in BAT through the upregulation of *UCP1* [24]. This action is potentiated by *PRDM16*, which may be a cross-talk regulator of BAT thermogenesis and glucose clearance in other tissues [25]. BAT is an important glucose-clearing organ, and a decreased expression of *Slc2a4* (GLUT4) may suggest impaired glucose clearing by BAT. Chronic treatment of DIO rats with AM630 also reduced *UCP1* and *Slc2a4* mRNA levels and resulted in a decrease in BAT mass.

Browning of WAT has been proposed as a mechanism to combat obesity and its co-morbidities [26], and browning is typically defined as an increase in the UCP1 mRNA/protein or an increase in other brown adipocyte genes, such as *CPT1b* and *PDRM16*. While the CB_2_ agonist JWH-133 increases the UCP1 protein in lean patient-derived white adipocytes, this effect is severely blunted in obese patient-derived white adipocytes [27]. In CB_2_-knockout mice fed an HFD, there is no evidence of browning of eWAT, as defined by a lack of changes in *UCP1*, *Cox8b*, or *Cidea* mRNA levels [8]. In our study, we observed genes involved in thermogenesis with decreased *CPT1b* mRNA in pWAT and *Adrb3* and *PRDM16* mRNA in eWAT following treatment with AM1241, while AM630 treatment decreased *Adrb3* and *PRDM16* mRNA levels in eWAT.

Interestingly, the heart weighed more in the AM630 and AM1241 treatment groups compared with vehicle-treated DIO rats; however, when expressed as a percentage of the total body weight, only with the CB_2_ agonist AM1241 did we observe a significant increase in the heart weight (grams). This result reinforces the notion that increasing CB_2_ may have a cardiac effect rather than a metabolic influence on these animals. Although we did not test the impact of CB_2_ agonist treatment on the cardiac tissue function, we have previously reported that these animals experienced a reduced systolic BP, urinary protein, urinary albumin, urinary sodium excretion, and renal fibrotic markers [6]. Therefore, the agonism of CB_2_ appears to be having a positive renal-cardiac effect, which is supported by the increase in the heart size. CB_2_ receptor agonist exposure in the heart has been found to protect the heart from ischemic damage in rats [28]. However, an enlarged heart can be a result of cardiomegaly, explaining the increased organ weight [29]. CB_2_ has been identified as having an anti-inflammatory effect during cardiac tissue remodeling [30]. This would be anticipated given the role of CB_2_ in cardiac repair, and we would have seen an increase in systemic cytokines if this were the case; however, no change was detected despite the increase in heart weight. Moreover, there is a need to determine macrophage infiltration (M1 and M2 markers) at variable time points in adipose tissues to determine if there are direct effects of modulating CB_2_ on inflammation. Given the close association between obesity and cardiovascular disease risk, further investigation of the effects of CB_2_ activation on the cardiac muscle is warranted. Despite the lack of effects, such as weight loss and changes in food intake, we demonstrated a yet-to-be-elucidated role in CB_2_ modulation in altering plasma leptin levels and marker of thermogenesis. However, this study had a number of limitations, namely, the dose and duration of treatment. At the time of the study design, few investigations into CB_2_ modulation effects had been conducted in DIO animal models, with no studies focusing on rodents; therefore, the concentrations of AM630 [31] and AM1241 [32] were based on non-obese models.

While further dose–response studies are warranted, the doses used here are similar to those used in other studies using AM630 [33] and AM1241 [34]. We hypothesized a link between CB_2_ modulation and cardiac-renal effects; while we present evidence of this at an in vivo level, we did not look at cellular adaptations in cardiac tissue. Similarly, the reduction in circulating leptin in plasma that occurred with 6 weeks of both CB_2_ agonism and antagonist looking at changes in central leptin signaling would be beneficial.

Overall, CB_2_ modulation in DIO had little effect on inflammatory markers in the plasma and in adipose tissue. However, CB_2_ modulation reduced the circulating leptin levels in both the CB_2_ agonist and antagonist treatment groups. Moreover, CB_2_ modulation altered specific adipocyte genes involved in thermogenic pathways. Collectively, our results show that CB_2_ modulation plays a role in decreasing circulating leptin levels and modulating thermogenic mRNA in adipose tissue.

## 4. Materials and Methods

### 4.1. Animals and Experimental Protocol

All animal experimental procedures were approved by the Howard Florey Animal Ethics Committee (AEC 11-036), which operates under the guidelines of the National Health and Medical Research Council of Australia. Seven-week-old male Sprague Dawley rats were used (The Animal Resource Centre, Canning Vale, WA, Australia). Following a 1–2-week acclimatization period, the rats were individually housed in a plastic tub with stainless steel lid (cage dimensions; width 27.5 × length 41.0 × height 25.5 cm) (R.E. Walters, Sunshine, Victoria, Australia) in an environmentally controlled laboratory (ambient temperature 22–24 °C) with a 12 h light/dark cycle (07:00–19:00).

### 4.2. Rodent Model of Diet-Induced Obesity and AM1241 and AM630 Pharmacological Treatment

Following the acclimatization period, rats received a high-fat diet (HFD) 21% fat content (equating to 40% digestible energy) from lipids (Specialty Feeds SF00-219, Glen Forrest, WA, Australia) for 9 weeks, as described in our previously published study [13]. Throughout the study, animals could access food and water ad libitum. Animals were then maintained on the HFD and treated for a further six weeks with a daily i.p. injection, with either vehicle (0.9% isotonic saline solution containing 0.75% Tween 80: n = 9–10), 3 mg/kg body weight of AM1241 (Cayman Chemicals, Ann Arbour, MI, USA), or 0.3 mg/kg body weight of AM630 (Cayman Chemicals, Ann Arbour, MI, USA) dissolved in the vehicle solution (n = 9–10). These compounds and their doses were chosen at the time of the initiation of the study due to the following papers: AM1241 [32] and AM630 [31]. EchoMRI Whole-Body Composition Analyzer (EchoMRI-900; EchoMRI, Houston, TX, USA) was used to determine body composition, as previously described [6].

Following treatment, rats were anesthetized with 3% isoflurane inhalation (Abbott Laboratories, Chicago, IL, USA) with cardiac blood collected to confirm their death; then, all other major organs including fat pads were removed post-mortem, weighed, snap frozen in liquid nitrogen, and stored at −80 °C for further analyses.

### 4.3. RNA Extraction

Adipose tissue fat pads were carefully removed and stored using previously described methods [13]. qPCR was performing on a LightCycler 480 (Roche, Millers Point, NSW, Australia), as follows: initial heating to 50 °C for 2 min, then 95 °C for 10 min, before each cycle consisted of 95 °C for 15 s and 60 °C for 2 min for 40 cycles, and then all samples were cooled to 25 °C. Samples of mRNA were amplified to test the expression of Brown adipocyte genes: UCP1, PRDM16, and CPT1B; WAT specific genes: HOXC9 and TCF21; receptors; CB_1_, CB_2_, and β_3_ adrenoceptor; transporters: GLUT1, GLUT4, IL-1β, IL-6, TNF-α, and adiponectin or leptin mRNA (Thermo Fisher Scientific, Waltham, MA, USA; Table 3). The data were normalized to hypoxanthine phosphoribosyltransferase 1 (HPRT1).

### 4.4. Plasma Hormone and Cytokine Analysis

Cardiac blood was extracted at the time of death and transferred to EDTA tubes (McFarlane Medical, Victoria, Australia), where it was processed as previously described [35]. Plasma samples were prepared following the manufacturer’s instructions for analysis of diabetes 5-plex panel and the rat cytokine 24-plex panel multiplex protein arrays (BioRad, BioRad Laboratories, Munich, Germany). The diabetes 5-plex panel consisted of the following measurements: 1. ghrelin; 2. leptin; 3. glucagon; 4. PAI-1 (*plasminogen activator inhibitor-1*); and 5. GLP-1 (*glucagon-like peptide-1*). The rat cytokine 24-plex panel kit consisted of the following measurements: 1. EPO (*erythropoietin*), 2. G-CSF (*granulocyte colony stimulating factor*), 3. GM-CSF (*granulocyte-macrophage colony stimulating factor*), 4. GRO/KC (*growth-related oncogene*), 5. IFN-γ *(interferon gamma*), 6. IL-α (*interleukin 1-alpha*), 7. IL-β (*interleukin 1-beta*), 8. IL-2 (*interleukin 2*), 9. IL-4 (*interleukin 4*), 10. IL-5 (*interleukin 5*), 11. IL-6 (*interleukin 6*), 12. IL-10 (*interleukin 10*), 13. IL-12p70 (*interleukin 12p70*), 14. IL-13. (*interleukin 13*), 15. IL-17α (*interleukin 17α*), 16. IL-18 (*interleukin 18*), 17. M-CSF (*macrophage colony-stimulating factor*), 18. MCP-1 (*monocyte chemotactic protein 1*), 19. MIP-3α (*macrophage inflammatory protein 3α*), 20. RANTES (*regulated on activated normal T-cells expressed and secreted*), 21. TNF-α (*tumor necrosis factor alpha*), and 22. VEGF (*vascular endothelial growth factor*). MIP-1α (*macrophage inflammatory protein 1α)* and IL-7 did not work for any sample and were thus excluded from the analysis. Plasma adiponectin was analyzed according to manufacturing instructions (AdipoGen, Liestal, Switzerland).

### 4.5. Statistical Analysis

Real-time semi-quantitative PCR (qPCR) values are presented as arbitrary units, mean ± SEM, normalized to housekeeping genes and expressed as 2^−ΔΔCT^ in arbitrary units. GraphPad 8.0 Prism software, all data are presented as mean ± SEM. Analysis of the groups were determined using an unpaired or paired *t*-test as appropriate; where data were not normally distributed, a non-parametric Mann–Whitney or Wilcoxon matched-pairs test was completed. Significance was accepted when *p* ≤ 0.05.

## 5. Conclusions

Following the investigation of the effects of both CB_2_ agonist AM1241 and CB_2_ antagonist AM630 on the inflammation and the metabolic adaptations in obesity, it is concluded that both treatments reduce circulating leptin in the absence of weight loss and alter the expression of genes that mediate the thermogenic response.

## Figures and Tables

**Figure 1 ijms-24-07601-f001:**
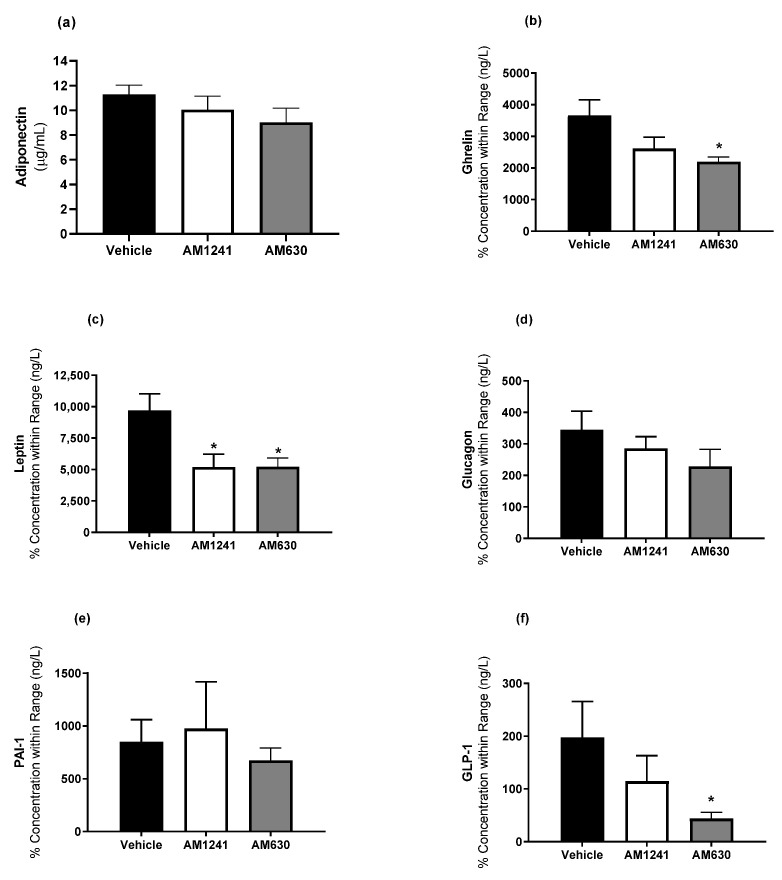
The effect of AM1241 or AM630 treatment in DIO rats on hormones involved in energy homeostasis. DIO rats were injected daily with AM1241 (3 mg/kg of body weight, ip), AM630 (0.3 mg/kg of body weight, ip), or vehicle for six weeks. Plasma (**a**) adiponectin, (**b**) ghrelin, (**c**) leptin, (**d**) glucagon, (**e**) PAI-1 or (**f**) GLP-1. All results are presented as a mean ± SEM from n = 8–9 (AM1241)-, n = 8–9 (AM630)-, and n = 8–9 (vehicle)-treated rats on an HFD (exception PAI n = 4, 6 and 7 for AM1241, AM630 and vehicle, respectively). Significance * *p* < 0.05 compared to vehicle.

**Figure 2 ijms-24-07601-f002:**
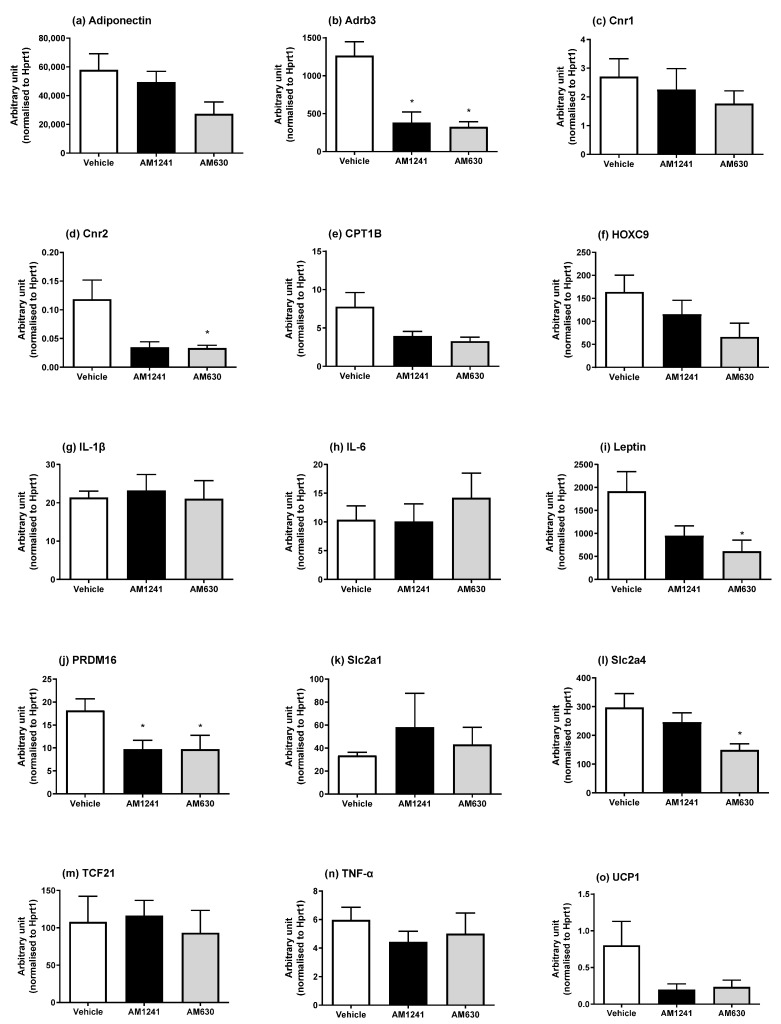
Effect of AM1241 or AM630 treatment on DIO rats’ epididymal WAT mRNA expression. DIO rats were injected daily with AM1241 (3 mg/kg of body weight, ip), AM630 (0.3 mg/kg of body weight, ip), or vehicle for six weeks. All results are presented as a mean ± SEM in arbitrary units (normalized to housekeeping gene *HPRT1*) from n = 6–7 (AM1241)-, n = 6–9 (AM630)-, and n = 7–9 (vehicle)-treated DIO rats. Significance * *p* < 0.05 compared to vehicle.

**Figure 3 ijms-24-07601-f003:**
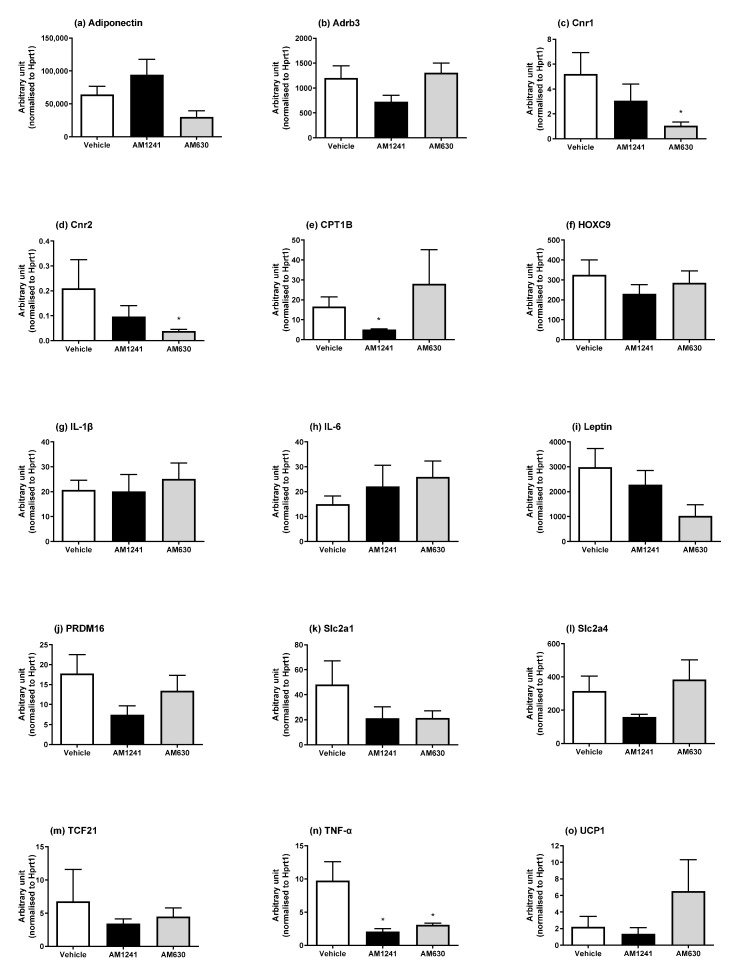
Effect of AM1241 or AM630 treatment on DIO rats peri-renal WAT mRNA expression. DIO rats were injected daily with either AM1241 (3 mg/kg of body weight, ip), AM630 (0.3 mg/kg body weight, ip), or vehicle for six weeks. All results are presented as a mean ± SEM in arbitrary units (normalized to housekeeping gene HPRT1) from n = 6–8 (AM1241)-, n = 6–10 (AM630)-, and n = 8–10 (vehicle)-treated rats on an HFD (exception n = 5 for *Slc2a1* and *Tcf21* in AM1241 and *UCP1* for AM630, n = 6 for *Tcf21* vehicle). Significance * *p* < 0.05 compared to vehicle.

**Figure 4 ijms-24-07601-f004:**
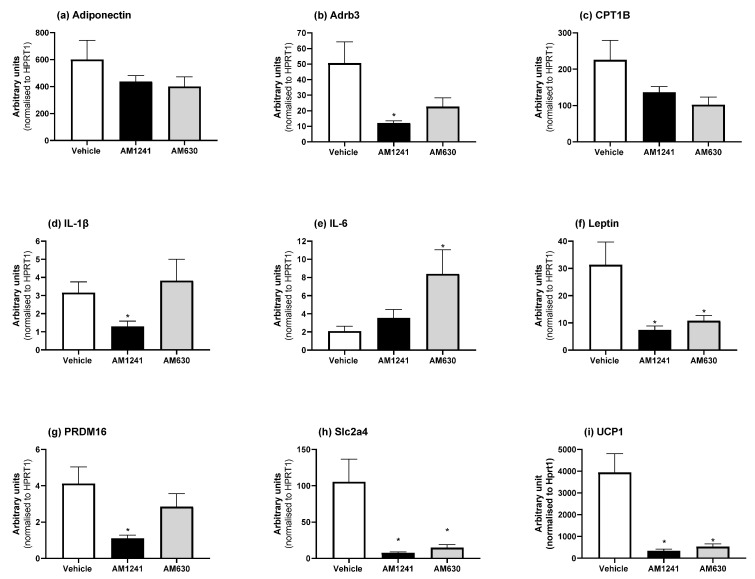
Effect of AM1241 or AM630 treatment in DIO rats on BAT mRNA expression. DIO rats were injected daily with either AM1241 (3 mg/kg of body weight, ip), AM630 (0.3 mg/kg of body weight, ip), or vehicle for six weeks. All results are presented as a mean ± SEM in arbitrary units (normalized to housekeeping gene HPRT1) from n = 7–9 (AM1241)-, n = 7–9 (AM630)-, and n = 7–9 (vehicle)-treated rats on an HFD (exception n = 6 for *IL-1β* for AM1241 and AM630). Significance * *p* < 0.05 compared to vehicle.

**Table 1 ijms-24-07601-t001:** The effect of AM1241 or AM630 treatment in DIO on tissue weights in grams (g) and % of total body weight (% bw). DIO rats were injected daily with AM1241 (3 mg/kg of body weight, ip), AM630 (0.3 mg/kg of body weight, ip), or vehicle for six weeks. All results are presented as a mean ± SEM from n = 8–9 (AM1241)-, n = 9–10 (AM630; exception heart n = 5)-, and n = 10 (vehicle; exception heart n = 6)-treated rats on an HFD. Significance * *p* < 0.05 compared to vehicle.

Tissue Weight	Vehicle	AM1241	AM630
Lean mass (g)	512.3 ± 13.5	553.0 ± 17.8	550.2 ± 10.7 *
Lean mass (% bw)	77.5 ± 1.1	80.0 ± 2.0	79.3 ± 1.3
Fat mass (g)	122.5 ± 10.6	117.7 ± 16.9	115.9 ± 11.4
Fat mass (% bw)	18.3 ± 1.1	16.6 ± 2.1	16.6 ± 1.5
Heart (g)	1.60 ± 0.08	1.86 ± 0.07 *	2.02 ± 0.12 *
Heart (% bw)	0.246 ± 0.01	0.266 ± 0.01	0.293 ± 0.019 *
Liver (g)	22.27 ± 0.84	24.26 ± 1.07	25.68 ± 1.46
Liver (% bw)	3.340 ± 0.088	3.633 ± 0.124	3.701 ± 0.155
Epididymal adipose tissue (g)	10.32 ± 0.77	8.11 ± 0.72	8.20 ± 0.58 *
Epididymal adipose tissue (% bw)	1.544 ± 0.103	1.146 ± 0.077 *	1.197 ± 0.082 *
Peri-renal adipose tissue (g)	11.90 ± 1.15	9.77 ± 0.96	13.10 ± 1.11
Peri-renal adipose tissue (% bw)	1.766 ± 0.138	1.384 ± 0.107	1.887 ± 0.140
Brown adipose tissue (g)	1.08 ± 0.10	0.91 ± 0.11	0.76 ± 0.10 *
Brown adipose tissue (% bw)	0.152 ± 0.008	0.129 ± 0.013	0.124 ± 0.018

**Table 2 ijms-24-07601-t002:** Plasma cytokines following AM1241 or AM630 treatment in DIO rats. DIO rats were injected daily with AM1241 (3 mg/kg of body weight, ip), AM630 (0.3.mg/kg of body weight, ip), or vehicle for six weeks. All results are presented as a mean ± SEM from n = 6–9 (AM1241)-, n = 6–9 (AM630)-, and n = 6–9 (vehicle)-treated rats on an HFD in % concentration within range (ng/L) (exception n = 5–7 in G-CSF and GM-CSF).

Cytokine	Vehicle	AM1241	AM630
EPO	583.5 ± 109	680 ± 174.3	786.8 ± 109.3
G-CSF	23.3 ± 6.1	34.7 ± 14.3	31.4 ± 14.5
GM-CSF	147.2 ± 55.7	200.7 ± 115.5	77.6 ± 22.9
GRO/KC	261.5 ± 87	183 ± 44.7	265.4 ± 40.5
IFN-γ	194.5 ± 46.8	225.2 ± 87.7	210.8 ± 60.4
IL-1α	154.2 ± 44.2	126.3 ± 34.1	95.8 ± 19.8
IL-1β	4098 ± 1179	2308 ± 628.7	2292 ± 685.7
IL-2	338.7 ± 66.4	353.9 ± 91.9	282.7 ± 52.3
IL-4	196.1 ± 63.2	113.1 ± 38	125.3 ± 33.5
IL-5	357.6 ± 69.4	270.3 ± 46	285.1 ± 49.1
IL-6	163.2 ± 73	127.3 ± 47.21	168.2 ± 50.7
IL-10	1161 ± 324.4	500.9 ± 145.6	631.5 ± 95.1
IL-12p70	235.4 ± 79.4	121.2 ± 38.1	182.6 ± 62.0
IL-13	102.4 ± 29.5	61.5 ± 17.5	60.1 ± 17.4
IL-17α	104.8 ± 27.4	80 ± 18.4	80.7 ± 18.5
IL-18	3430 ± 703.7	3236 ± 645.8	4012 ± 438.2
MCSF	477.8 ± 23	495.6 ± 58.7	419.5 ± 23.0
MCP-1	957.4 ± 151.4	1210 ± 241.7	881.9 ± 144.1
MIP-3α	105.7 ± 27.2	68.2 ± 15.8	74.6 ± 10.4
RANTES	296.2 ± 70.1	370.2 ± 63.1	299.5 ± 45.2
TNF-α	155.9 ± 51.7	157.7 ± 66.2	104.9 ± 25.3
VEGF	51.1 ± 14.8	32.5 ± 7.6	43.8 ± 12.3

**Table 3 ijms-24-07601-t003:** Rat TaqMan gene expression assay—adipose.

Genes	Exon Boundary	Taqman Catalogue Number	Amplicon Length
*Adiponectin*	2–3	Rn00595250_m1	63
*Adrb3 (β3-AR)*	2–3	Rn01478698_g1	131
*Cnr1 (CB* _1_ *)*	1–2	Rn00562880_m1	81
*Cnr2 (CB* _2_ *)*	1–2	Rn01637601_m1	68
*CPT1B*	11–12	Rn00682395_m1	83
*HOXC9*	1–2	Rn01532842_m1	94
*HPRT1*	8–9	Rn01527840_m1	64
*IL-1β*	5–6	Rn00580432_m1	74
*IL-6*	3–4	Rn01410330_m1	121
*Leptin*	1–2	Rn00565158_m1	92
*PRDM16*	5–6	Rn01516224_m1	65
*Slc2a1 (GLUT 1)*	8–9	Rn01417099_m1	73
*Slc2a4 (GLUT 4)*	9–10	Rn00562597_m1	75
*TCF21*	1–2	Rn01537344_m1	95
*TNF-α*	2–3	Rn99999017_m1	108
*UCP1*	2–3	Rn00562126_m1	69

*Adrb3 (β3-AR)*: adrenoceptor beta 3, *Cnr1 (CB*_1_*)*: cannabinoid receptor 1, *Cnr2 (CB*_2_*)*: cannabinoid receptor 2, *CPT1B*: carnitine palmitoyltransferase 1B, *HOXC9*: homeobox C9, *HPRT1*: hypoxanthine phosphoribosyltransferase 1, *IL-1β*: Interleukin 1 beta, *IL-6*: Interleukin 6, *PRDM16*: PR/SET domain 16, *Slc2a1* (GLUT 1): glucose transporter 1, *Slc2a4* (GLUT 4): glucose transporter 4, *TCF21*: transcription factor 21, *TNF-α*: tumor necrosis factor alpha, *UCP1*: uncoupling protein 1.

## Data Availability

The data presented in this study are available on request from the corresponding author.

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
