# Peer review of "Treatment of Diet-Induced Obese Rats with CB2 Agonist AM1241 or CB2 Antagonist AM630 Reduces Leptin and Alters Thermogenic mRNA in Adipose Tissue"

_ijms, 2023, doi:10.3390/ijms24087601_

Round 1

Reviewer 1 Report

O’Keefe et al aimed to examine the effects of pharmacological modulation of the cannabinoid type 2 receptor (CB2) on inflammation and adaptations to the obese state in a diet-induced obesity (DIO) model. The role of CB2 receptor in obesity is not clear, with conflicting evidence reported in the literature. Whereas some studies show that CB2 agonism can reduce body weight and inflammation, others report no effect or even contradicting results. There is therefore a gap in knowledge due to the inconsistency in the reported effects of CB2 activation or inhibition on obesity-related outcomes across different studies.

In this study, the authors employed an established animal model to investigate the molecular mechanisms involved in CB2 agonism and CB2 antagonism treatment in adipose tissue. Interestingly, the results demonstrate that CB2 agonists and antagonists reduce circulating leptin levels without impacting body weight, food intake, or liver weight. Furthermore, both treatments modulate mRNA levels responsible for thermogenesis, suggesting their potential as therapeutic options for reducing inflammation and co-morbidities associated with obesity.

The following comments need to be addressed before accepting the paper for publication:

1. The data show that there is no effect on body weight with AM1241 or AM630 treatment, but there was a decrease in adipose tissue weight. This reviewer wonders if the increased liver and heart weights compensate for the loss body weight. Additionally, since CB2 agonism has been shown to increase bone mass (PMID: 16407142), it would be interesting to know if there is any evidence for changes in bone/muscle (lean) weights following the treatment.

2. What is the explanation for the increase in heart and liver mass following treatment with AM1241 or AM630?

3. Reduced leptin levels, likely due to reduced adipose tissue mass, may lead to increased leptin sensitivity, which in turn can inhibit food intake and lower body weight. Although none of these systemic effects were found in this study, it is still important to report changes in leptin homeostasis. Have the authors measured the effect of the treatment on central leptin sensitivity, such as hypothalamic STAT3 phosphorylation.

4. Despite the reduction in fat mass following treatment with AM1241 or AM630, there was an unexpected decrease in Adrb3 expression in these tissues, which could mediate changes in leptin levels and fat mass. The authors could assess the protein expression of Adrb3 or measure norepinephrine levels in adipose tissues treated with the drugs to investigate this further.

5. Since different transcriptional data were found between the three fat pads (epididymal, peri-renal, and BAT), can the author assess the same set of genes in subcutaneous fat pat too?

6. Moreover, the authors report a significant decreased UCP1 expression in BAT, which should reduce thermogenesis and energy expenditure. May this effect explain the unaffected body weight loss following the treatment?

7. If neither CB1 nor CB2 receptors are expressed in BAT, what could explain the transcriptional changes measured in BAT following the treatment?  The data may suggest off-target effects of the drugs or indirect effects of the drugs on norepinephrine levels. This should either tested experimentally or better discussed.

Author Response

1. The data show that there is no effect on body weight with AM1241 or AM630 treatment, but there was a decrease in adipose tissue weight. This reviewer wonders if the increased liver and heart weights compensate for the loss body weight. Additionally, since CB2 agonism has been shown to increase bone mass (PMID: 16407142), it would be interesting to know if there is any evidence for changes in bone/muscle (lean) weights following the treatment.

While there was a significant increase in heart weight (~0.3 to 0.4 g) there were no significant changes in liver weight and these small changes are unlikely to have compensated for any changes in body weight in the animals which weighed ~680 g on average. While no change in food intake was also observed we did assess body composition via Echo MRI and have now added this data to Table 1 Page 3. AM630 treatment resulted in an increase in lean mass in grams but not as a % of total body weight. Whereas no alteration in total fat mass as measured by the Echo MRI were detected. 

2. What is the explanation for the increase in heart and liver mass following treatment with AM1241 or AM630?

The observed changes in heart weight are discussed on Page 11, lines 4-20. While an enlarged heart could be a sign of cardiomegaly, we also observed a decrease in blood pressure in these animals1 as well as no alterations in circulating inflammatory markers and thus it appears that these treatments did not have a negative effect. However further investigation into cardiac function (such as Fibrotic markers, ischemic markers2, and atrial function3) are warranted in future research into these compounds. Given our results we hypothesis that CB2 modulation is having a direct cardio effect and warrants investigation.

1.     https://www.ncbi.nlm.nih.gov/pmc/articles/PMC5338152/

2.     https://pubmed.ncbi.nlm.nih.gov/27614871/

3.     https://pubmed.ncbi.nlm.nih.gov/30500553/

There was a trend for an increase in liver weight, which was not significant.

3. Reduced leptin levels, likely due to reduced adipose tissue mass, may lead to increased leptin sensitivity, which in turn can inhibit food intake and lower body weight. Although none of these systemic effects were found in this study, it is still important to report changes in leptin homeostasis. Have the authors measured the effect of the treatment on central leptin sensitivity, such as hypothalamic STAT3 phosphorylation.

We did not investigate the brain as the primary focus of study was peripheral effects. Future studies could investigate any effect of our treatments on central leptin sensitivity.

4. Despite the reduction in fat mass following treatment with AM1241 or AM630, there was an unexpected decrease in Adrb3 expression in these tissues, which could mediate changes in leptin levels and fat mass. The authors could assess the protein expression of Adrb3 or measure norepinephrine levels in adipose tissues treated with the drugs to investigate this further.

Antibodies against GPCRs are notoriously unreliable and unvalidated for specificity as highlighted previously in a review1.

While the authors have used a β3-AR antibody from Santa Cruz Biotechnology previously2 which showed selectivity as assessed by using tissues obtained from β3-AR knockout mice, this antibody is no longer commercially available, and we are unable to test the selectivity of new β3-AR antibodies as we do not have access to β3-AR knockout mice currently. The authors have attempted in the past to establish methods to measure β3-AR protein levels in adipose tissue using radioligand binding approaches with [125I]-cyanopindolol but have been unsuccessful due to the high content of lipids in the adipose tissue (the radioligand is highly lipophilic for instance). The mouse β3-AR in primary brown adipocytes is regulated at the level of mRNA3 as opposed to similar GPCRs such as the β1 or β2-AR which are regulated by GRK and β-arrestin mediated desensitisation mechanisms. Hence measurement of Adrb3 mRNA levels should reflect protein levels of the β3-AR in adipose tissue.

1.     https://link.springer.com/article/10.1007/s00210-009-0395-y

2.     https://pubmed.ncbi.nlm.nih.gov/15665039/

3.     https://pubmed.ncbi.nlm.nih.gov/8969197/

5. Since different transcriptional data were found between the three fat pads (epididymal, peri-renal, and BAT), can the author assess the same set of genes in subcutaneous fat pat too?

We did not collect the subcutaneous fat pads as it was not the focus at the time of the study design. The fat pads analysed are classified are both traditional white and ‘brite’ depots1. Allowing the detection of ‘browning’ effects of CB2 modulation in WAT and thermogenesis in BAT.

1.     https://pubmed.ncbi.nlm.nih.gov/21828341/

6. Moreover, the authors report a significant decreased UCP1 expression in BAT, which should reduce thermogenesis and energy expenditure. May this effect explain the unaffected body weight loss following the treatment?

This has now been added in text: Page 10 lines 19 it now reads - in BAT, may suggest a blunting of thermogenic responses in these animals which may at least partially explain the lack of alteration in body weight observed in these animals.

7. If neither CB1 nor CB2 receptors are expressed in BAT, what could explain the transcriptional changes measured in BAT following the treatment?  The data may suggest off-target effects of the drugs or indirect effects of the drugs on norepinephrine levels. This should either tested experimentally or better discussed.

Cnr1 and Cnr2 mRNA was not detected in BAT, hence the effects of AM630 and AM1241 on the transcriptional changes in BAT are most likely due to a secondary indirect effect1. Since BAT is highly innervated, this could be due to alterations in the sympathetic tone to BAT. Another possible mechanism is the reduction in circulating leptin levels following both AM630 and AM1241 treatment. Leptin administration to rats increases BAT Ucp1 mRNA levels which is dependent upon sympathetic activation of BAT. This has now been added in text: Page 10 lines 6-12.

1.     https://pubmed.ncbi.nlm.nih.gov/9688627/

Reviewer 2 Report

The authors have investigated the potential therapeutic site of manipulating CB2 receptor during HFD-induced obesity, by testing a lot representative markers like body weight, leptin, UCP-1 level (just name few) from different adipose tissue. These works are remarkable with great insights on further studies. Here are few comments that I have for this study:

The authors have clearly described the contradicted findings regarding CB2 agonism usage in HFD animal models, and well explained the potential factors lead to different outcomes of this study with previous study. Regarding the similar trend of effects from the one agonist and one antagonist used in this study, it is reasonable to give a more detailed explanation of why the authors, in this study, prefer AM1241 over the other forms of agonists.

The treatment of CB2 agonist/antagonist last for 6 weeks, and the result 2.1 together with table 1 well descripted the animal organ weight / fat gain level. Did the authors also test the agonist/antagonist effects during these 6 weeks as a time-dependent way? This will sacrifice more animal but could demonstrate the drug effect curves during the treatment period. Similar question for results 2.2 and figure 1, did this study test these factors on the date after 6 weeks treatment while sacrificed the animal? The author could take blood from animal weekly, for instance, to see how these tested factors change according to CB2 agonist/antagonist treatment.

The authors can have a better description of how to calculate of the two different dosage of the agonist and antagonist, respectively (e.g. EC50/IC50, peak blood concentration, effective drug dosage regarding expression level of CB2 on interested organ/tissue...).

In page 9 of 14, line 17-20, based on the authors’ statement, it is better to discuss the details of the comparable between 6 weeks AM1241 treatment vs. 15 days JWH-015 treatment towards the mRNA level of TNF-α (e.g. EC50, pharmacokinetics...)?

The last question is: does animal show tolerance to the CB2 agonist/antagonist used in this study among HFD-induced obesity mechanism? Or if there were withdraw effects at the time after the 6 weeks CB2 agonist/antagonist treatment? If the answer is yes, I will not be surprised that there is a possibility of the presented data underestimates the CB2 agonist/antagonist effect by missing the plateau/maximum drug effect.

Author Response

The authors have clearly described the contradicted findings regarding CB2 agonism usage in HFD animal models, and well explained the potential factors lead to different outcomes of this study with previous study. Regarding the similar trend of effects from the one agonist and one antagonist used in this study, it is reasonable to give a more detailed explanation of why the authors, in this study, prefer AM1241 over the other forms of agonists.

There are indeed several different agonists and antagonists that are available that target CB2. These compounds and their dose were chosen at the time of the initiation of the study due to the following papers

AM630 - https://www.sciencedirect.com/science/article/abs/pii/S0899900714002974

AM1241 - https://pubmed.ncbi.nlm.nih.gov/21810593/

This has now been added in text: Page 11, lines 40-41.

The treatment of CB2 agonist/antagonist last for 6 weeks, and the result 2.1 together with table 1 well descripted the animal organ weight / fat gain level. Did the authors also test the agonist/antagonist effects during these 6 weeks as a time-dependent way? This will sacrifice more animal but could demonstrate the drug effect curves during the treatment period. Similar question for results 2.2 and figure 1, did this study test these factors on the date after 6 weeks treatment while sacrificed the animal? The author could take blood from animal weekly, for instance, to see how these tested factors change according to CB2 agonist/antagonist treatment.

Glucose tolerance and insulin sensitivity were completed at baseline and towards the end of the treatment period via tail snip1. Regular tail snips were unable to be undertaken outside of these due to the impact on the animals. While we cannot rule out if changes in mRNA or plasma markers or tissue weights would have varied if we had animals that underwent shorter or longer treatments, this is common to all animal studies, and it is well beyond the scope of this manuscript to repeat all this data in a full time-course study. We also believe that additional studies are against the Australian code for the care and use of animals for scientific purposes , per the Australian NHMRC2

1.     https://pubmed.ncbi.nlm.nih.gov/25537025/

2.     https://www.nhmrc.gov.au/research-policy/ethics/animal-ethics/3rs

The authors can have a better description of how to calculate of the two different dosages of the agonist and antagonist, respectively (e.g. EC50/IC50, peak blood concentration, effective drug dosage regarding expression level of CB2 on interested organ/tissue...).

Thank you for your comment, there is very little information available on pharmacokinetic (PK) data for both drugs. Moreover, there are no PK experiments on rats, however, we do have this information at the mouse receipt for the potency of the drugs.

These concentrations remain relevant with recent publications testing AM630 1 mg/kg (https://www.mdpi.com/1422-0067/24/4/3828) and AM1241 at an identical concentration as utilised in our current study of 3 mg/kg in rats https://www.ncbi.nlm.nih.gov/pmc/articles/PMC6454175/

This has now been added in text: Page 11, lines 2-6.

In page 9 of 14, line 17-20, based on the authors’ statement, it is better to discuss the details of the comparable between 6 weeks AM1241 treatment vs. 15 days JWH-015 treatment towards the mRNA level of TNF-α (e.g. EC50, pharmacokinetics...)?

We have previously speculated that the differences observed may be due to differences in doses or duration of the studies. It is unclear as to what the reviewer is further asking us to elaborate on further.

The last question is: does animal show tolerance to the CB2 agonist/antagonist used in this study among HFD-induced obesity mechanism? Or if there were withdraw effects at the time after the 6 weeks CB2 agonist/antagonist treatment? If the answer is yes, I will not be surprised that there is a possibility of the presented data underestimates the CB2 agonist/antagonist effect by missing the plateau/maximum drug effect.

We did not test for the withdrawal effects of the compounds. CB modulation have been found to have a transient decrease in food intake at the onset of treatment1. We assessed body composition via Echo MRI and have added this data to Table 1 Page 3. Given we detected no change in weight or food throughout the intervention period, it is unlikely that following the withdrawal effects would have yielded further significant changes. We of course cannot rule out if we treated the animals for longer or at higher (or lower) doses that this may have resulted in different outcomes. These limitations have been added to page 11 lines 1-13.

1.     https://pubmed.ncbi.nlm.nih.gov/36232744/

Reviewer 3 Report

In the manuscript entitled” Treatment of diet-induced obese rats with CB2 2 agonist AM1241 or CB2 antagonist AM630 3 reduces leptin and alters thermogenic mRNA 4 in adipose tissue,” authors investigate the effect of  CB2 agonism and CB2 antagonism treatment on some molecular mechanisms involved in inflammation and adipose tissue browning in a diet-induced obesity model. The question posed by the authors is so interesting. The results are original. However, some issues should be considered before the final decision. Here, you can constructively find my comments:

1-It seems data regarding FBS, and TG circulating levels can provide valuable information on the HFD-induced model. It is highly recommended that the authors provide the above-mentioned data.

2- Results: authors stated that “ We have previously demonstrated that AM1241 and AM630 treatment in DIO rats has no impact on glucose tolerance or insulin tolerance [6].”  Here, they refer to previous work in 2016. I wonder if the authors used the samples from the aforementioned study. If yes authors should point it to the present manuscript? Moreover, they should provide information regarding the storage of samples from 2006 up to now. Otherwise, it is highly recommended that authors provide data on glucose tolerance or insulin tolerance

3- Inflammation is a well-established hallmark of obesity. Investigation of anti-inflammatory cytokines such as IL-10, markers of macrophage infiltration, M1 markers, and M2 markers in adipose tissue can provide helpful information regarding the possible effects of modulating CB2 with pharmacological 24 treatments on inflammation.

4- In the results, authors stated that “which is important 7 for adipocyte thermogenesis and lipolysis, as well as expression of the Cnr1 and Cnr2 8 receptors.” As for lipolysis markers authors only measure CPT1 transcript levels. The measurement of only one marker cannot be sufficient for evaluating lipolysis in adipose tissue. Please remove this word or evaluate other markers.

5- A major point seems to be a discussion. The discussion is partly a result. The authors reported a considerable result. However, they did not have a general view of the results. Finally, they should provide how we can deduce from the results.

6-Authors also should points to study limitations in the discussion.

7-It is highly recommended that authors provide a graphical abstract to allow readers to quickly gain an understanding of the take-home message of the paper.

8-In methods and Results: Authors stated that both pre-renal WAT and epidydimal WAT has been used for some experiments. I wonder why the authors used two fat depots. They should discuss the possible reasons for different results in pre-renal and epidydimal WAT .

 In the manuscript entitled” Treatment of diet-induced obese rats with CB2 2 agonist AM1241 or CB2 antagonist AM630 3 reduces leptin and alters thermogenic mRNA 4 in adipose tissue,” authors investigate the effect of  CB2 agonism and CB2 antagonism treatment on some molecular mechanisms involved in inflammation and adipose tissue browning in a diet-induced obesity model. The question posed by the authors is so interesting. The results are original. However, some issues should be considered before the final decision. Here, you can constructively find my comments:

1-It seems data regarding FBS, and TG circulating levels can provide valuable information on the HFD-induced model. It is highly recommended that the authors provide the above-mentioned data.

2- Results: authors stated that “ We have previously demonstrated that AM1241 and AM630 treatment in DIO rats has no impact on glucose tolerance or insulin tolerance [6].”  Here, they refer to previous work in 2016. I wonder if the authors used the samples from the aforementioned study. If yes authors should point it to the present manuscript? Moreover, they should provide information regarding the storage of samples from 2006 up to now. Otherwise, it is highly recommended that authors provide data on glucose tolerance or insulin tolerance

3- Inflammation is a well-established hallmark of obesity. Investigation of anti-inflammatory cytokines such as IL-10, markers of macrophage infiltration, M1 markers, and M2 markers in adipose tissue can provide helpful information regarding the possible effects of modulating CB2 with pharmacological 24 treatments on inflammation.

4- In the results, authors stated that “which is important 7 for adipocyte thermogenesis and lipolysis, as well as expression of the Cnr1 and Cnr2 8 receptors.” As for lipolysis markers authors only measure CPT1 transcript levels. The measurement of only one marker cannot be sufficient for evaluating lipolysis in adipose tissue. Please remove this word or evaluate other markers.

5- A major point seems to be a discussion. The discussion is partly a result. The authors reported a considerable result. However, they did not have a general view of the results. Finally, they should provide how we can deduce from the results.

6-Authors also should points to study limitations in the discussion.

7-It is highly recommended that authors provide a graphical abstract to allow readers to quickly gain an understanding of the take-home message of the paper.

8-In methods and Results: Authors stated that both pre-renal WAT and epidydimal WAT has been used for some experiments. I wonder why the authors used two fat depots. They should discuss the possible reasons for different results in pre-renal and epidydimal WAT .

 In the manuscript entitled” Treatment of diet-induced obese rats with CB2 2 agonist AM1241 or CB2 antagonist AM630 3 reduces leptin and alters thermogenic mRNA 4 in adipose tissue,” authors investigate the effect of  CB2 agonism and CB2 antagonism treatment on some molecular mechanisms involved in inflammation and adipose tissue browning in a diet-induced obesity model. The question posed by the authors is so interesting. The results are original. However, some issues should be considered before the final decision. Here, you can constructively find my comments:

1-It seems data regarding FBS, and TG circulating levels can provide valuable information on the HFD-induced model. It is highly recommended that the authors provide the above-mentioned data.

2- Results: authors stated that “ We have previously demonstrated that AM1241 and AM630 treatment in DIO rats has no impact on glucose tolerance or insulin tolerance [6].”  Here, they refer to previous work in 2016. I wonder if the authors used the samples from the aforementioned study. If yes authors should point it to the present manuscript? Moreover, they should provide information regarding the storage of samples from 2006 up to now. Otherwise, it is highly recommended that authors provide data on glucose tolerance or insulin tolerance

3- Inflammation is a well-established hallmark of obesity. Investigation of anti-inflammatory cytokines such as IL-10, markers of macrophage infiltration, M1 markers, and M2 markers in adipose tissue can provide helpful information regarding the possible effects of modulating CB2 with pharmacological 24 treatments on inflammation.

4- In the results, authors stated that “which is important 7 for adipocyte thermogenesis and lipolysis, as well as expression of the Cnr1 and Cnr2 8 receptors.” As for lipolysis markers authors only measure CPT1 transcript levels. The measurement of only one marker cannot be sufficient for evaluating lipolysis in adipose tissue. Please remove this word or evaluate other markers.

5- A major point seems to be a discussion. The discussion is partly a result. The authors reported a considerable result. However, they did not have a general view of the results. Finally, they should provide how we can deduce from the results.

6-Authors also should points to study limitations in the discussion.

7-It is highly recommended that authors provide a graphical abstract to allow readers to quickly gain an understanding of the take-home message of the paper.

8-In methods and Results: Authors stated that both pre-renal WAT and epidydimal WAT has been used for some experiments. I wonder why the authors used two fat depots. They should discuss the possible reasons for different results in pre-renal and epidydimal WAT .

 In the manuscript entitled” Treatment of diet-induced obese rats with CB2 2 agonist AM1241 or CB2 antagonist AM630 3 reduces leptin and alters thermogenic mRNA 4 in adipose tissue,” authors investigate the effect of  CB2 agonism and CB2 antagonism treatment on some molecular mechanisms involved in inflammation and adipose tissue browning in a diet-induced obesity model. The question posed by the authors is so interesting. The results are original. However, some issues should be considered before the final decision. Here, you can constructively find my comments:

1-It seems data regarding FBS, and TG circulating levels can provide valuable information on the HFD-induced model. It is highly recommended that the authors provide the above-mentioned data.

2- Results: authors stated that “ We have previously demonstrated that AM1241 and AM630 treatment in DIO rats has no impact on glucose tolerance or insulin tolerance [6].”  Here, they refer to previous work in 2016. I wonder if the authors used the samples from the aforementioned study. If yes authors should point it to the present manuscript? Moreover, they should provide information regarding the storage of samples from 2006 up to now. Otherwise, it is highly recommended that authors provide data on glucose tolerance or insulin tolerance

3- Inflammation is a well-established hallmark of obesity. Investigation of anti-inflammatory cytokines such as IL-10, markers of macrophage infiltration, M1 markers, and M2 markers in adipose tissue can provide helpful information regarding the possible effects of modulating CB2 with pharmacological 24 treatments on inflammation.

4- In the results, authors stated that “which is important 7 for adipocyte thermogenesis and lipolysis, as well as expression of the Cnr1 and Cnr2 8 receptors.” As for lipolysis markers authors only measure CPT1 transcript levels. The measurement of only one marker cannot be sufficient for evaluating lipolysis in adipose tissue. Please remove this word or evaluate other markers.

5- A major point seems to be a discussion. The discussion is partly a result. The authors reported a considerable result. However, they did not have a general view of the results. Finally, they should provide how we can deduce from the results.

6-Authors also should points to study limitations in the discussion.

7-It is highly recommended that authors provide a graphical abstract to allow readers to quickly gain an understanding of the take-home message of the paper.

8-In methods and Results: Authors stated that both pre-renal WAT and epidydimal WAT has been used for some experiments. I wonder why the authors used two fat depots. They should discuss the possible reasons for different results in pre-renal and epidydimal WAT .

 In the manuscript entitled” Treatment of diet-induced obese rats with CB2 2 agonist AM1241 or CB2 antagonist AM630 3 reduces leptin and alters thermogenic mRNA 4 in adipose tissue,” authors investigate the effect of  CB2 agonism and CB2 antagonism treatment on some molecular mechanisms involved in inflammation and adipose tissue browning in a diet-induced obesity model. The question posed by the authors is so interesting. The results are original. However, some issues should be considered before the final decision. Here, you can constructively find my comments:

1-It seems data regarding FBS, and TG circulating levels can provide valuable information on the HFD-induced model. It is highly recommended that the authors provide the above-mentioned data.

2- Results: authors stated that “ We have previously demonstrated that AM1241 and AM630 treatment in DIO rats has no impact on glucose tolerance or insulin tolerance [6].”  Here, they refer to previous work in 2016. I wonder if the authors used the samples from the aforementioned study. If yes authors should point it to the present manuscript? Moreover, they should provide information regarding the storage of samples from 2006 up to now. Otherwise, it is highly recommended that authors provide data on glucose tolerance or insulin tolerance

3- Inflammation is a well-established hallmark of obesity. Investigation of anti-inflammatory cytokines such as IL-10, markers of macrophage infiltration, M1 markers, and M2 markers in adipose tissue can provide helpful information regarding the possible effects of modulating CB2 with pharmacological 24 treatments on inflammation.

4- In the results, authors stated that “which is important 7 for adipocyte thermogenesis and lipolysis, as well as expression of the Cnr1 and Cnr2 8 receptors.” As for lipolysis markers authors only measure CPT1 transcript levels. The measurement of only one marker cannot be sufficient for evaluating lipolysis in adipose tissue. Please remove this word or evaluate other markers.

5- A major point seems to be a discussion. The discussion is partly a result. The authors reported a considerable result. However, they did not have a general view of the results. Finally, they should provide how we can deduce from the results.

6-Authors also should points to study limitations in the discussion.

7-It is highly recommended that authors provide a graphical abstract to allow readers to quickly gain an understanding of the take-home message of the paper.

8-In methods and Results: Authors stated that both pre-renal WAT and epidydimal WAT has been used for some experiments. I wonder why the authors used two fat depots. They should discuss the possible reasons for different results in pre-renal and epidydimal WAT .

Author Response

1-It seems data regarding FBS, and TG circulating levels can provide valuable information on the HFD-induced model. It is highly recommended that the authors provide the above-mentioned data.

The animals were fed a high-fat diet for 15 weeks, we assessed body composition via Echo MRI and added this data to Table 1, Page 3. The animals were fed a high-fat diet (HFD) 21% fat content (equating to 40% digestible energy) from lipids ad libitum. This is a commonly used diet that replicates ‘The Western Diet’1 . The serum triglycerides levels in rats consuming this diet have been published previosly1. We did not feel that the addition of plasma TG levels fits within the focus of this manuscript or adds considerably to it.

1.     https://pubmed.ncbi.nlm.nih.gov/22820146/

2- Results: authors stated that “ We have previously demonstrated that AM1241 and AM630 treatment in DIO rats has no impact on glucose tolerance or insulin tolerance [6].”  Here, they refer to previous work in 2016. I wonder if the authors used the samples from the aforementioned study. If yes authors should point it to the present manuscript? Moreover, they should provide information regarding the storage of samples from 2006 up to now. Otherwise, it is highly recommended that authors provide data on glucose tolerance or insulin tolerance

The animal study was conducted in 2015, with samples of tissues frozen in liquid nitrogen and then stored at -80 oC until required, plasma and serum were analysed in the same year and mRNA was completed in 2016 and 2017. The first author’s movement into a teaching-focused role within the University limited capacity to complete and finalise statistical data analysis and write-up of this manuscript. The timeline was also severely disrupted due to the COVID pandemic.

3- Inflammation is a well-established hallmark of obesity. Investigation of anti-inflammatory cytokines such as IL-10, markers of macrophage infiltration, M1 markers, and M2 markers in adipose tissue can provide helpful information regarding the possible effects of modulating CB2 with pharmacological 24 treatments on inflammation.

It was difficult to test every inflammatory cytokine possible. Hence, we decided to use the BioRad, Bio-Plex 24-plex array which can measure 24 different cytokine levels from the same sample simultaneously. This array was chosen to gain the most amount of information on a range of cytokines in a timely and cost-efficient manner. We believe that the data we present significantly advances the field and the targeting of CB2 receptors with these agonists/antagonists in a DIO rodent model.

4- In the results, authors stated that “which is important  for adipocyte thermogenesis and lipolysis, as well as expression of the Cnr1 and Cnr2 8 receptors.” As for lipolysis markers authors only measure CPT1 transcript levels. The measurement of only one marker cannot be sufficient for evaluating lipolysis in adipose tissue. Please remove this word or evaluate other markers.

Thank you for your comment, we have removed reference to lipolysis in this sentence.

5- A major point seems to be a discussion. The discussion is partly a result. The authors reported a considerable result. However, they did not have a general view of the results. Finally, they should provide how we can deduce from the results.

Further discussion has been added to fully emphasise the take-home message for the reader, rather than just the results. This now reads:

Page 10, lines 52-54 - Despite the lack of effects such as weight loss and changes in food intake, we demonstrated a yet-to-be-elucidated role in CB2 modulation in altering plasma leptin levels and marker of thermogenesis.

Page 11, lines 15-17- Collectively, our results show CB2 modulation plays a role in decreasing circulating leptin levels and modulating thermogenic mRNA in adipose tissue.  

6-Authors also should points to study limitations in the discussion.

Thank you, a discussion of limitations has now been included on page 11 lines 1-11.

7-It is highly recommended that authors provide a graphical abstract to allow readers to quickly gain an understanding of the take-home message of the paper.

Thank you for identifying this, a graphical abstract has now been included.

b8-In methods and Results: Authors stated that both pre-renal WAT and epidydimal WAT has been used for some experiments. I wonder why the authors used two fat depots. They should discuss the possible reasons for different results in pre-renal and epidydimal WAT

We collected epididymal fat as a traditional white fat pad and per-renal fat pad as a depot that has the potential to become more beige in response to stimuli1. There are many reasons why the gene expression is different in each fat pad, however, differences in gene expression are known to occur in different fat pads2. The differences were anticipated, and discussions focusing on the variations are beyond the scope of the current study.

1.     https://pubmed.ncbi.nlm.nih.gov/21828341/

2.     https://www.karger.com/Article/PDF/257511

Round 2

Reviewer 1 Report

The measurments of body composition by EchoMRI should be added to the Method section too. 

Except this, I have no further comments.

Author Response

The measurements of body composition by EchoMRI should be added to the Method section too. 

This has been added to the methods section, Page 11, lines 43-45.

Reviewer 2 Report

According to the authors’ responses to the round 1 review comments, the treatment time of AM630 applied in current study (6 weeks) is different from the literature that authors provided (https://www.sciencedirect.com/science/article/abs/pii/S0899900714002974, which was within 24 hour); for AM1241, the treatment time is also different from the paper provided (https://pubmed.ncbi.nlm.nih.gov/21810593/, which was 14 weeks). Another literature has been provided by authors (https://www.ncbi.nlm.nih.gov/pmc/articles/PMC6454175/) to support the usage of AM1241 with same dosage as in the current study, but the treatment time was only half of that in this study.

Authors have performed this study without sufficient pharmacokinetic data (which is admitted by the authors with respond “…there is very little information available on pharmacokinetic (PK) data for both drugs. Moreover, there are no PK experiments on rats…”).

By responding “…changes in mRNA or plasma markers or tissue weights would have varied if we had animals that underwent shorter or longer treatments, this is common to all animal studies…”, the authors fully aware of the necessary of well-designed experimental protocol(s) when testing chemical effect on mRNA/bio-markers/tissue weight.

I highly encourage the authors to perform pharmacokinetic studies, see if that data support the current dosages + treatment time (in order to catch the peak drug effects). Then can be more confident to conclude that an agonist and an antagonist, which generally thought to have oppose effects, come up with same effects in regulating leptin and altering thermogenic mRNA in adipose tissue. Also, authors may find more exciting evidence regarding the food intake / body weight during short period treatment (like 2.0h~48.0h) vs long period treatment (as current study).

Author Response

According to the authors’ responses to the round 1 review comments, the treatment time of AM630 applied in current study (6 weeks) is different from the literature that authors provided (https://www.sciencedirect.com/science/article/abs/pii/S0899900714002974, which was within 24 hour); for AM1241, the treatment time is also different from the paper provided (https://pubmed.ncbi.nlm.nih.gov/21810593/, which was 14 weeks). Another literature has been provided by authors (https://www.ncbi.nlm.nih.gov/pmc/articles/PMC6454175/) to support the usage of AM1241 with same dosage as in the current study, but the treatment time was only half of that in this study.

Authors have performed this study without sufficient pharmacokinetic data (which is admitted by the authors with respond “…there is very little information available on pharmacokinetic (PK) data for both drugs. Moreover, there are no PK experiments on rats…”).

By responding “…changes in mRNA or plasma markers or tissue weights would have varied if we had animals that underwent shorter or longer treatments, this is common to all animal studies…”, the authors fully aware of the necessary of well-designed experimental protocol(s) when testing chemical effect on mRNA/bio-markers/tissue weight.

I highly encourage the authors to perform pharmacokinetic studies, see if that data support the current dosages + treatment time (in order to catch the peak drug effects). Then can be more confident to conclude that an agonist and an antagonist, which generally thought to have oppose effects, come up with same effects in regulating leptin and altering thermogenic mRNA in adipose tissue. Also, authors may find more exciting evidence regarding the food intake / body weight during short period treatment (like 2.0h~48.0h) vs long period treatment (as current study).

Thank you, similar studies have been published without prior pharmacokinetic studies using AM1241 or AM630 in similar concentrations and doses to ours1,2,3. 

We did measure food intake and weight daily in the animals in the current study (data shows; there was no difference because of the treatment), moreover, it is highly unlikely that any detectable changes in body composition in a DIO rat model would occur within a 24–48-hour period. The focus of this study was not on the acute effects of these drugs on the large amount of data presented, but rather on the more longer-term effects. We cannot rule out, however, that the various markers measured in the study would be different at different times of the day, or if treatment periods were shorter or longer. This is a common limitation in ALL research and thus all you can do is control what you can and clearly document when and how you have collected the samples. We believe that we have adequately done this within the manuscript to allow the readers to repeat our experiments or to further investigate these compounds to add to the knowledge base.

1.     https://pubmed.ncbi.nlm.nih.gov/11514083/

2.     https://pubmed.ncbi.nlm.nih.gov/19027877/

3.     https://pubmed.ncbi.nlm.nih.gov/21838753/

Reviewer 3 Report

Authors respond to most of the comments, however, some issues still need more consideration:

1-I cannot see a graphical abstract provided in the manuscript.

2-Investigation of markers of macrophage infiltration, M1 markers, and M2 markers in adipose tissue can provide helpful information regarding the possible effects of modulating CB2 with pharmacological 24 treatments on inflammation. Authors should include it in the study limitation.

Author Response

1-I cannot see a graphical abstract provided in the manuscript.

Thank you, the graphical abstract was included with the response to the last comments. It has been attached in this response. 

2-Investigation of markers of macrophage infiltration, M1 markers, and M2 markers in adipose tissue can provide helpful information regarding the possible effects of modulating CB2 with pharmacological 24 treatments on inflammation. Authors should include it in the study limitation.

Reference to this has now been included on Page 10, lines 50-52.